# Bioactivity of alginetin, a caramelization product of pectin: Cytometric analysis of rat thymic lymphocytes using fluorescent probes

**Sayaka Doi[1]☯, Mina Kawamura[2]☯, Keisuke Oyama[3], Tetsuya Akamatsu[2], Mizuki Mizobuchi[4], Yasuo Oyama[2], Toshiya Masuda[1], Norio Kamemura[4]\***

**1** Faculty of Human Life Science, Osaka City University, Osaka City, Osaka, Japan, **2** Faculty of Bioscience and Bioindustry, Tokushima University, Tokushima City, Tokushima Japan, **3** Surgery Division, Sakai City Medical Center, Sakai City, Osaka, Japan, **4** Department of Food-Nutritional Sciences, Faculty of Life Sciences, Tokushima Bunri University, Tokushima, Japan

☯ These authors contributed equally to this work.
\* kamemura.norio@tks.bunri-u.ac.jp

**Data Availability Statement:** All relevant data are within the manuscript.

**Funding:** The Japan Society for the Promotion of Science (Tokyo, Japan) supports this study with

## Abstract

Alginetin is the major product formed from pentoses and hexurionic acids. Alginetin is produced by cooking process of food including pection, a naturally-occurring polysacharride found in many plants. However, the biological interaction and toxicity of alginetin are not known at all. The aim of the present study was to investigate the cellular actions of alginetin on rat thymic lymphocytes. The effects of alginetin on the cell were examined using flow cytometry with fluorescent probes. Alginetin increased cellular content of non-protein thiols ($[NPT]_i$) and elevated intracellular $Zn^{2+}$ levels ($[Zn^{2+}]_i$). Chelation of intracellular $Zn^{2+}$ reduced the effect of alginetin on $[NPT]_i$, and chelation of external $Zn^{2+}$ almost completely diminished alginetin-induced elevation of $[Zn^{2+}]_i$, indicating that alginetin treatment increased $Zn^{2+}$ influx. Increased $[NPT]_i$ and $[Zn^{2+}]_i$ levels in response to alginetin were positively correlated. Alginetin protected cells against oxidative stress induced by hydrogen peroxide and $Ca^{2+}$ overload by calcium ionophore. It is considered that the increases in $[NPT]_i$ and $[Zn^{2+}]_i$ are responsible for the cytoprotective activity of alginetin because NPT attenuates oxidative stress and $Zn^{2+}$ competes with $Ca^{2+}$. Alginetin may be produced during manufacturing of jam, which may provide additional health benefits of jam.

## Introduction

Alginetin (3,8-dihydroxy-2-methylchromone) may be a forgotten compound, as nearly all papers concerning this molecule were published 65–84 years ago [1]. Moreover, there have been no reports of pharmacological and/or toxicological actions of alginetin. Therefore, our study is probably the first investigation of alginetin bioactivity.

Pectin is a natural produced essential ingredient in preserves. Pectin is a type of starch, called a heteropolysaccharide, that occurs naturally in the cell walls of fruits and vegetables and gives them structure. Alginetin (3,8-dihydroxy-2-methylchromone) was first obtained by heating alginic acid with water in an autoclave and was also obtained from pectin and gummic

Grant-in-Aids for Scientific Research (C26340039, 18K14408).

**Competing interests:** The authors have declared that no competing interests exist.

acid in the 1930's [1]. Alginetin is a hydrolysis degradation product of pectin [2] Pectin is used as a food additive and occurs naturally in most fruits. Pectin is responsible for giving jellies their gel-like consistency, resulting in better spreadability. Use of a pressure cooker to make jam from fruits may result in alginetin production. Although bioactivity of alginetin has not been studied, the chemical structure suggests that the compound possesses antioxidant activity similar to other methylchromone derivatives with hydroxyl groups that have antioxidative properties [3]. In this study, we examined bioactivity of alginetin in rat thymic lymphocytes using flow cytometry with fluorescent probes. Beneficial cellular activity of alginetin would provide additional value for foods in which alginetin is produced during the manufacturing process.

## Materials and methods

### Preparation of alginetin

$KH_2PO_4$ and $Na_2HPO_4$ (9:1, 50 g) were added to a methanolic solution (50 mL) of D-glucuronolactone (1 g), then mixed well. After removal of methanol *in vacuo*, residual solids were heated at 130˚C for 1 h under $N_2$. The resulting brown solid was extracted twice with methanol (50 mL), and the combined extract was evaporated completely, resulting in a dark brown residue (210 mg). The residue was purified by chromatography using an octadecyl silane silica gel column with a stepwise gradient using 20%, 30%, and 40% aqueous methanol containing 1% acetic acid, and 100% methanol containing 1% acetic acid (each 120 mL). Eluate was collected in 60 mL fractions and the third fraction containing alginetin (17 mg) was evaporated *in vacuo*. These chemicals were purchased from Wako Pure Chemicals (Osaka, Japan). The structure of the isolated alginetin was characterized by nuclear magnetic resonance spectroscopy (NMR) as follows; 1H-NMR (400 MHz, CD3OD): δ 7.61 (1H, d, J = 7.6 Hz), 7.26 (1H, t, J = 7.6 Hz), 7.19 (1H, d, J = 7.6 Hz), 2.55 (3H, s). NMR spectra were obtained from a JNM-ECZ400S (400 MHz, JEOL, Tokyo, Japan).

### Fluorescent probes and chemicals

Propidium iodide (PI) was used to assess cell death. Estimation of cellular content of nonprotein thiols was performed with 5-chloromethylfluorescein diacetate (5-CMF-DA) [4]. Fluo-Zin-3-AM™ was used to study changes in intracellular $Zn^{2+}$ levels ($[Zn^{2+}]_i$) [5]. These probes were commercially obtained from Invitrogen and Thermo Fisher Scientific Inc. (Eugene, Oregon, USA). Membrane-impermeable and -permeable $Zn^{2+}$ chelators, diethylenetriamine-N,N, N',N'',N''-pentaacetic acid and N,N,N',N'-tetrakis (2-pyridylmethyl)ethylenediamine (DTPA and TPEN, respectively), were purchased from Dojin Chemicals (Kumamoto, Japan). Other chemicals were products of Wako Pure Chemicals (Osaka, Japan).

### Cell suspension

The use of experimental animals in this study was approved by the committee of Tokushima University and Tokushima Bunri University (Tokushima, Japan) (Registration number in Tokushima university: T29-52, Registration number in Tokushima Bunri university: H30-3).

Cell suspensions were prepared from thymus glands as follows. Briefly, thymus glands were dissected from 8- to 12-week-old male Wistar rats (Total number of rat; n = 8, 280–340 g / Charles River Laboratories Japan, Kanagawa, Japan) anesthetized with 50 mg/kg i.p. thiopental sodium (Ravonal™, Mitsubishi Tanabe Pharma, Osaka, Japan). The glands were sliced with a razor. The slices were triturated in chilled Tyrode's solution (150 mM NaCl, 5 mM KCl, 2 mM $CaCl_2$, 1 mM $MgCl_2$, 5 mM glucose) to dissociate individual lymphocytes. The pH of solution

was adjusted at 7.3–7.4 with 5 mM 4-(2-hydroxyethyl)-1-piperazineethanesulfonic acid and appropriate amount of NaOH. The solution containing dissociated cells was passed through a mesh (diameter: 25 μm) to prepare the cell suspension (approximately $5 \times 10^5$ cells/mL). Cells were incubated at 36–37°C for 1 h before experimental use. The cell morphology images of the cells cultured in the present of alginetin were performed by a inverted microscope (Nikon Eclipse TS100, Nikon,Tokyo, Japan).

## Fluorescence measurements of cellular parameters

Alginetin, 5-CMF-DA, FluoZin-3-AM, DTPA, and TPEN were initially dissolved in dimethyl sulfoxide (DMSO). DMSO was present at concentrations of 0.1–0.3% in final conditions, which did not induce cell death. Cellular and membrane parameters were measured using fluorescent probes and a flow cytometer equipped with a software package for data collection and analysis (CytoACE-150; JASCO, Tokyo, Japan). The excitation wavelength used for the fluorescent probes was 488 nm, and emission was detected at 530 ± 15 nm for 5-CMF and FluoZin-3 and at 600 ± 20 nm for PI. PI, which stains dead cells, was added to cell suspensions at a final concentration of 5 μM. 5-CMF and FluoZin-3 fluorescence were measured only in cells exhibiting no PI fluorescence (living cells with intact cell membranes). To estimate changes in $[NPT]_i$, primarily glutathione, cells were incubated with 1 μM 5-CMF-DA for 30 min before fluorescence analysis. The correlation coefficient between flow cytometric and biochemical determination of glutathione was 0.965 [4]. To monitor changes in $[Zn^{2+}]_i$, cells were incubated with 500 nM FluoZin-3-AM for 60 min before fluorescence measurements were conducted.

## Experimental protocol

Alginetin (3–100 mM in 2 μL of DMSO) was added to cell suspensions (1.998 mL per one test tube) and the mixtures were incubated at 36–37°C. Each cell suspension (100 μL) was analyzed using flow cytometry to assess alginetin-induced changes in cellular parameters. Fluorescence data acquisition from $3 \times 10^3$ cells required 10–15 s. In our previous study to examine the cytotoxicity of $H_2O_2$ as oxidative stress and A23187 (Calcimycin), a divalent cation ionophore (Sigma-Aldrich Co, St. Louis, MO, USA) as $Ca^{2+}$ overload [6], the cells were incubated with $H_2O_2$ or A23187 for 3–4 h to induce cell death in 20–40% of cells. Change in $[NPT]_i$ by alginetin was examined 2 h after the application in order to suggest the [NPT]i before the occurrence of cell death induced by $H_2O_2$.

## Statistical analysis

Statistical analysis was performed using Excel Toukei 2010 (Social Survey Research Information Co., Ltd. Tokyo, Japan). Statistical analysis was performed using ANOVA with post-hoc Tukey's multivariate analysis. P-value < 0.05 was considered statistically significant. Data reported in this study were mean ± standard deviation of 4–8 experiments.

# Results

## Changes in 5-CMF fluorescence in response to alginetin

The histogram representing distribution of 5-CMF fluorescence in cells incubated with 100 μM alginetin for 2 h was shifted to a higher fluorescence intensity compared with the control histogram (Fig 1A) demonstrating that alginetin treatment increased $[NPT]_i$. Alginetin augmented 5-CMF fluorescence at 10 μM and significant augmentation was observed across the range of 30–100 μM alginetin (Fig 1B). Moreover, alginetin was changed the cell

(A)

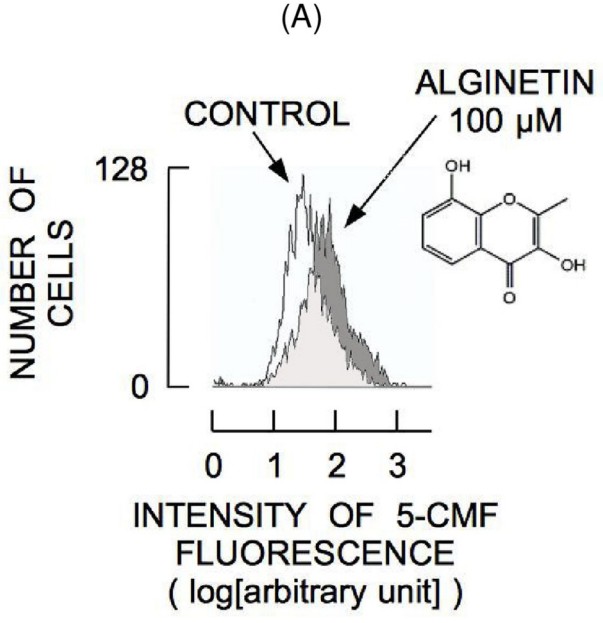

(B)

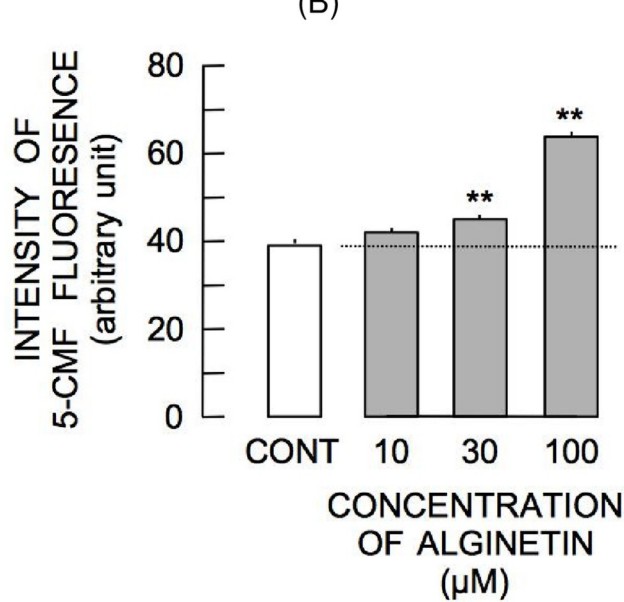

(C)

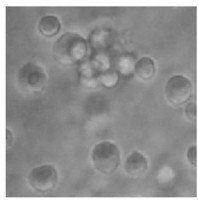

**Fig 1. Augmentation of 5-CMF fluorescence by alginetin.** (A) Shift of 5-CMF fluorescence histogram in response to 2 h alginetin treatment. The inset shows the chemical structure of alginetin. (B) Concentration-dependent augmentation of mean intensity of 5-CMF fluorescence induced by alginetin. Asterisks (**) indicate significant differences (P < 0.01) in 5-CMF fluorescence intensity between the control group (CONT) and the groups of cells treated with 30–100 μM alginetin. (C) The cell morphology images in response to 2 h alginetin treatment.

morphology on rat thymic lymphocytes (Fig 1C). A previous study demonstrated that elevation of $[Zn^{2+}]_i$ precedes increases in $[NPT]_i$ [7]. Therefore, to determine if $Zn^{2+}$ contributed to increased $[NPT]_i$, the effect of 100 μM alginetin was evaluated in the presence of 10 μM TPEN, an intracellular $Zn^{2+}$ chelator. As shown in Fig 2, incubation of cells with TPEN for 2 h reduced steady state intensity of 5-CMF fluorescence. Furthermore, TPEN significantly reduced alginetin-induced 5-CMF, indicating involvement of intracellular $Zn^{2+}$ in alginetin-induced elevation of $[NPT]_i$.

## Change in FluoZin-3 fluorescence in response to alginetin

To determine whether alginetin increases $[Zn^{2+}]_i$, alginetin-induced changes in $[Zn^{2+}]_i$ were measured. As shown in Fig 3A, the histogram of FluoZin-3 fluorescence shifted to a higher intensity 1 h after treatment with 100 μm alginetin. Treatment with 30–100 μM alginetin for 1 h significantly increased the intensity of FluoZin-3 fluorescence (Fig 3B), indicating that alginetin induced elevation of $[Zn^{2+}]_i$. The observed changes in FluoZin-3 fluorescence induced by alginetin were positively correlated with those of alginetin-induced 5-CMF fluorescence (Fig 4).

Increases in $[Zn^{2+}]_i$ can occur due to $Zn^{2+}$ influx from the extracellular environment and/or intracellular $Zn^{2+}$ release. To determine the source of alginetin-induced elevation of $[Zn^{2+}]_i$, changes in FluoZin-3 fluorescence induced by 100 μM alginetin were examined in the presence of 10 μM DTPA, an external $Zn^{2+}$ chelator. Removal of external $Zn^{2+}$ by DTPA significantly reduced steady state intensity of FluoZin-3 fluorescence and significantly attenuated alginetin-induced augmentation of FluoZin-3 fluorescence (Fig 5). Thus, alginetin-induced increases in $[Zn^{2+}]_i$ were dependent on extracellular $Zn^{2+}$.

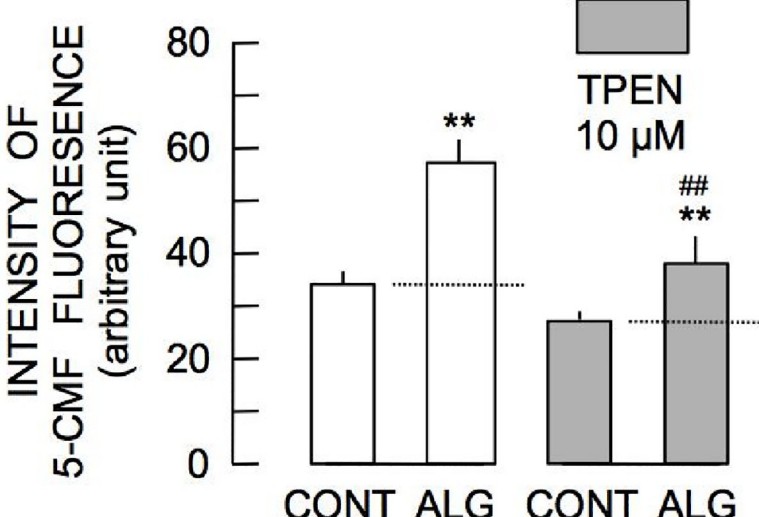

**Fig 2. Augmentation of 5-CMF fluorescence by alginetin in the absence and presence of TPEN.** Asterisks (**) indicate significant differences (P < 0.01) in 5-CMF fluorescence intensity between the control group (CONT) and the groups of cells treated with 100 μM alginetin (ALG) in the absence (left pair / open column) and presence (right pair / filled column) of 10 μM TPEN. Pound signs (##) show significant difference (P < 0.01) between the groups of cells treated with alginetin.

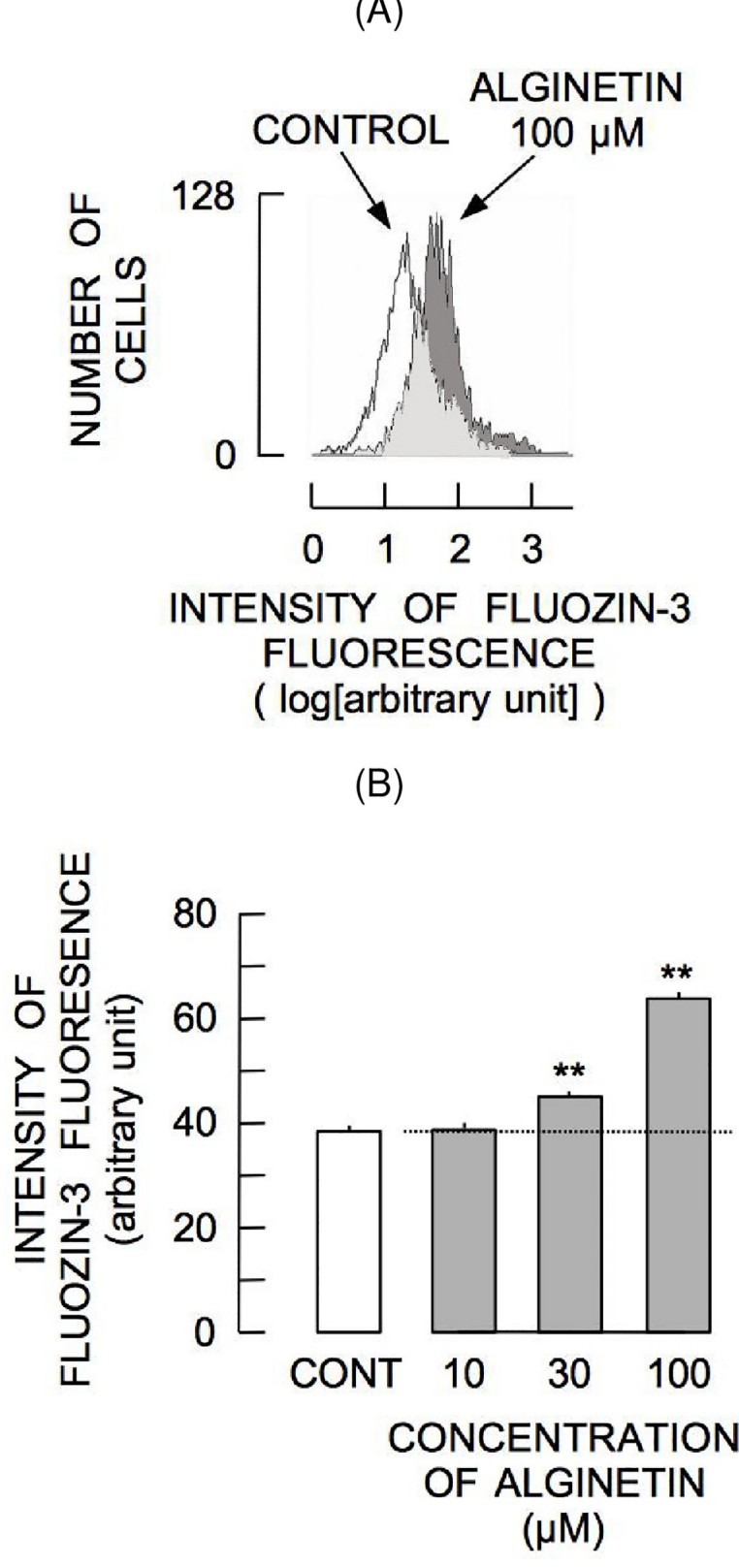

**Fig 3. Augmentation of FluoZin-3 fluorescence by alginetin.** (A) Shift of the FluoZin-3 fluorescence histogram in response to 1 h alginetin treatment. (B) Concentration-dependent augmentation of mean intensity of FluoZin-3

fluorescence by alginetin. Asterisks (**) indicate significant differences (P < 0.01) in FluoZin-3 fluorescence intensity between the control group (CONT) and the groups of cells treated with 30–100 μM alginetin.

### Cytoprotective actions of alginetin

Glutathione protects cells against oxidative stress [8, 9]. Therefore, alginetin-induced increases in $[NPT]_i$ may result in resistance to $H_2O_2$ toxicity. The effects of 10–100 μM alginetin on changes in cell lethality induced by 300 μM $H_2O_2$ were evaluated. As shown in Fig 6, incubation of cells with $H_2O_2$ for 4 h increased PI fluorescence, indicating increased cell death, while treatment with 100 μM alginetin for 4 h did not result in increased PI fluorescence. Cell death induced by $H_2O_2$ was significantly attenuated by simultaneous treatment with alginetin in a concentration-dependent manner (10–100 μM alginetin) (Fig 7A).

Treatment of cells with $H_2O_2$ increases intracellular $Ca^{2+}$ concentration ($[Ca^{2+}]_i$) [10], and sustained elevation of $[Ca^{2+}]_i$ is linked to cell death [11]. Furthermore, $Zn^{2+}$ partially attenuates $Ca^{2+}$-dependent cell death [12]. As alginetin treatment elevated $[Zn^{2+}]_i$, the possibility that alginetin treatment could protect against cell death induced by the calcium ionophore, A23187, was evaluated. As shown in Fig 7B, incubation of cells with 100 nM A23187 for 3 h significantly increased cell lethality. Simultaneous treatment with 30–100 μM alginetin significantly attenuated this effect.

## Discussion

### Cytoprotective actions of alginetin

Our study demonstrated that alginetin protects cells against oxidative stress induced by $H_2O_2$ and $Ca^{2+}$ overload induced by A23187. It is generally recognized that oxidative stress and $Ca^{2+}$ overload trigger cell death [13, 11]. Therefore, alginetin may be cytoprotective against many damaging stimuli. Alginetin, a degradation product of pectin, may be produced during jam

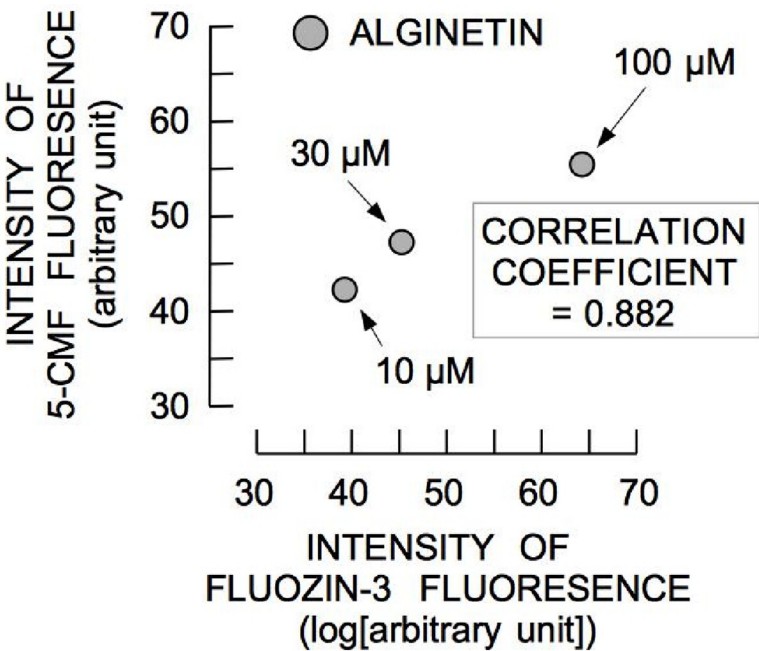

**Fig 4. Correlation between mean intensities of 5-CMF and FluoZin-3 fluorescence in cells treated with 10–100 μM alginetin for 1 h.**

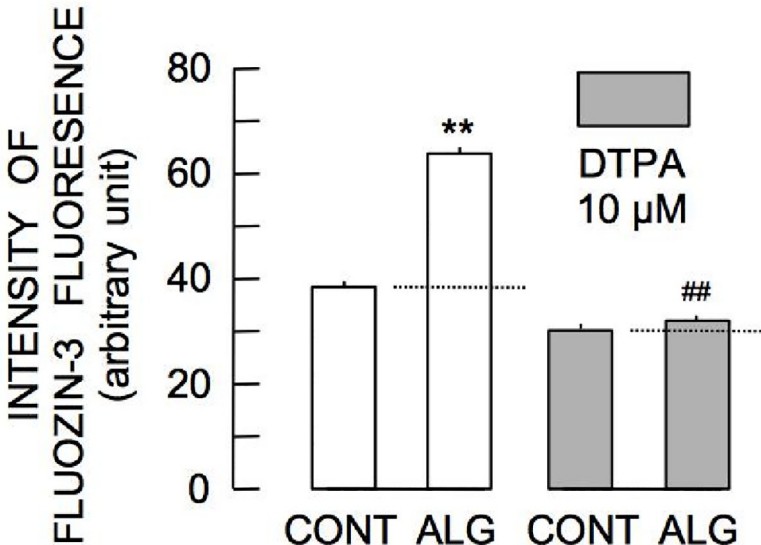

**Fig 5. Augmentation of FluoZin-3 fluorescence by alginetin in the absence and presence of DTPA.** Asterisks (**) indicate significant differences ($P < 0.01$) in FluoZin-3 fluorescence intensity between the control group (CONT) and groups of cells treated with 100 μM alginetin (ALG) for 1 h in the absence (left pair / open column) and presence (right pair / filled column) of 10 μM DTPA. Pound signs (##) indicate significant differences ($P < 0.01$) between the groups of cells treated with alginetin.

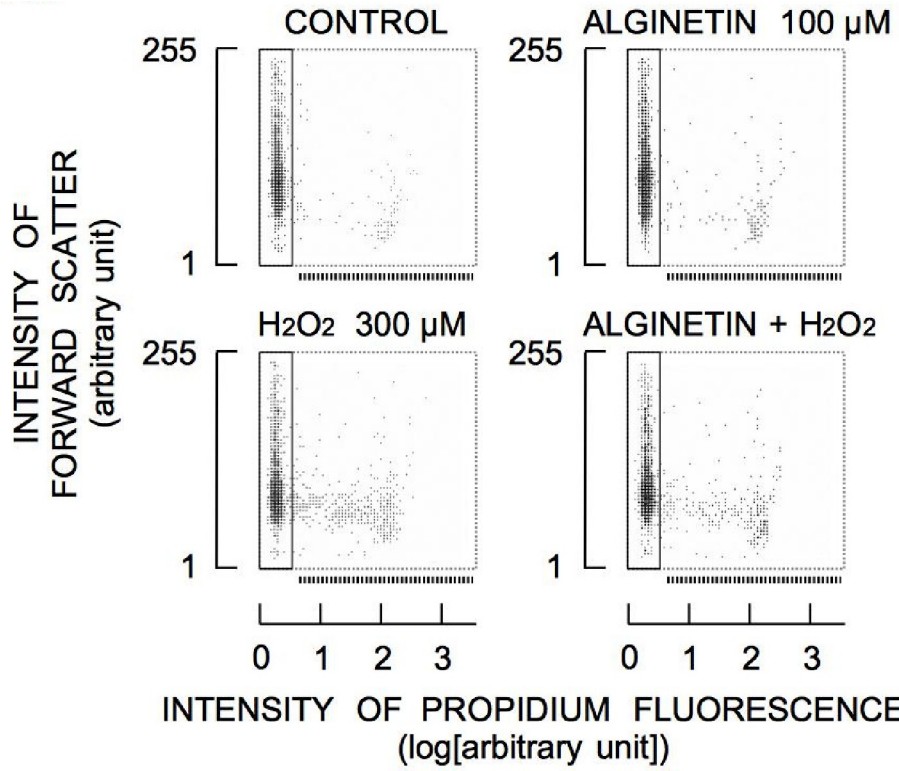

**Fig 6. Changes in the population of PI-stained cells treated with alginetin, $H_2O_2$, or both.** The dotted line under the cytogram indicates the region of cells exhibiting PI fluorescence.

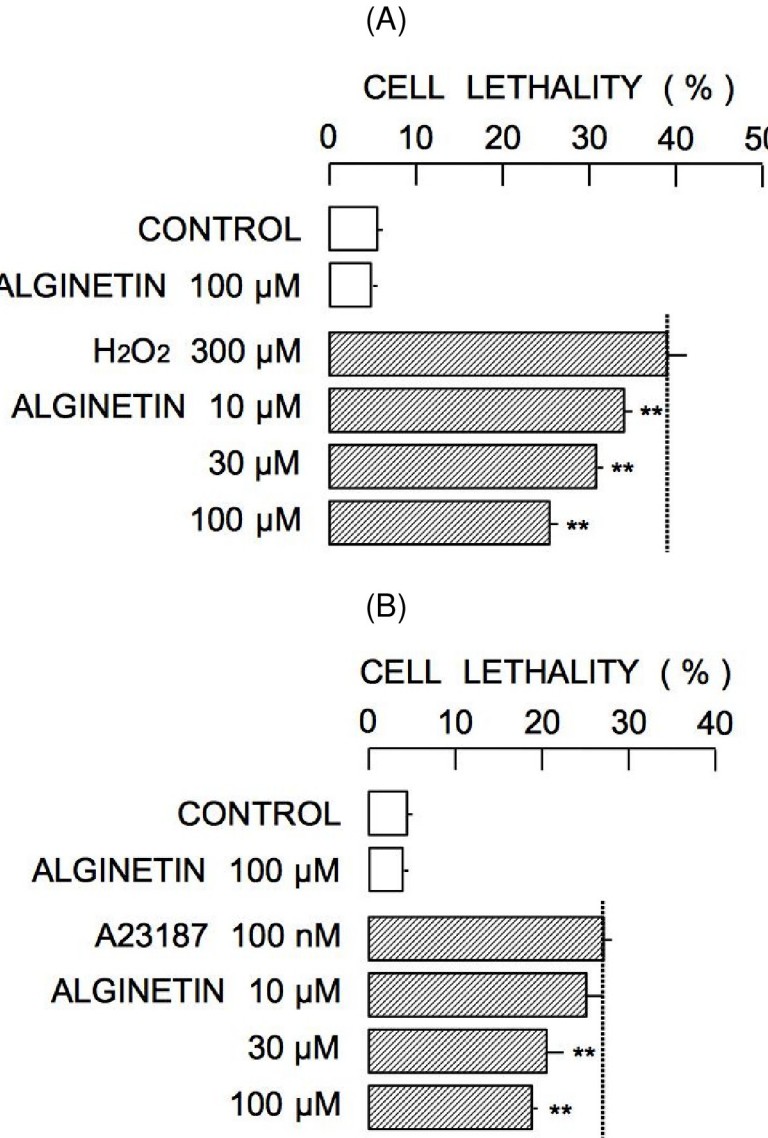

**Fig 7.** Protective effects of alginetin against oxidative stress induced by $H_2O_2$ (A) and $Ca^{2+}$ overload by A23187 (B). Asterisks (**) show significant differences between the groups of cells under insult-induced stress in the absence and presence of 10–100 μM alginetin.

production [1, 2]. Development of a process for jam production which produces alginetin may be attractive to health-oriented individuals.

## Cellular actions of alginetin

$[NPT]_i$ is required to maintain protein thiols in a reduced state and to support a variety of redox reactions for reducing ROS, detoxifying xenobiotics, and facilitating cell signaling. However, excessive oxidative stress results in indiscriminate and irreversible oxidation of protein thiols, depletion of $[NPT]_i$ and cell death [14]. Recently, Oyama et al. showed that the excessive of $[Zn^{2+}]_i$ increased $[NPT]_i$ [15]. Our results showed that alginetin elevated intracellular $Zn^{2+}$ levels and increased cellular content of non-protein thiols. Moreover, treatment with TPEN did not completely attenuate alginetin-induced increases in $[NPT]_i$. Alginetin treatment also

resulted in increased $[NPT]_i$ in the presence of DTPA, which almost completely suppressed alginetin-induced elevation of $[Zn^{2+}]_i$. These phenomena indicate that alginetin induced $Zn^{2+}$-dependent and -independent increases in $[NPT]_i$. $Zn^{2+}$ increases glutathione synthesis through an ARE-Nrf2–dependent pathway [8]. $Zn^{2+}$-independent mechanisms of action of alginetin will be evaluated in future studies. Increased $[NPT]_i$ in response to alginetin may protect against oxidative stress because $[NPT]_i$ is important in preventing pathological changes resulting from increased levels of reactive oxygen species [16]. A previous study demonstrated that removal of intracellular $Zn^{2+}$ by TPEN increased cytotoxicity of A23187, a calcium ionophore that causes $Ca^{2+}$ overload [12]. $Zn^{2+}$ competes with $Ca^{2+}$ at calcium binding proteins [17]. Therefore, increased $[Zn^{2+}]_i$ in response to alginetin may reduce $Ca^{2+}$ binding-related cell death, resulting in protection against $Ca^{2+}$ overload. Elevation of $[Zn^{2+}]_i$ in response to alginetin was extracellular $Zn^{2+}$-dependent because augmentation of FluoZin-3 fluorescence by alginetin was almost completely attenuated by cotreatment with DPTA. $Zn^{2+}$ influx is regulated by many zinc transporters such as ZIP4, 5, 6, 10, and 14 [18]. Alginetin may activate zinc transporters or increase membrane $Zn^{2+}$ permeability via zinc transporter-independent mechanisms.

## Conclusions

The biological interaction and toxicity of alginetin has not been studied in detail. Our study showed that alginetin increased cellular content of non-protein thiols and elevated intracellular $Zn^{2+}$ levels on rat thymic lymphocyte. These results indicate that alginetin increases cell vulnerability to oxidative stress on rat thymocytes. This study provides that alginetin, possessing cytoprotective activity, would provide additional health benefits of jam if it is produced from fruit pectin during jam manufacturing.

## Acknowledgments

We thank Kazumi Ishidoh for skillful assistance.

## Author Contributions

**Conceptualization:** Norio Kamemura.

**Data curation:** Sayaka Doi, Mina Kawamura.

**Formal analysis:** Sayaka Doi, Mina Kawamura, Keisuke Oyama.

**Investigation:** Sayaka Doi, Mina Kawamura, Keisuke Oyama, Tetsuya Akamatsu, Mizuki Mizobuchi, Yasuo Oyama, Norio Kamemura.

**Project administration:** Toshiya Masuda, Norio Kamemura.

**Supervision:** Yasuo Oyama, Toshiya Masuda.

**Visualization:** Yasuo Oyama.

**Writing – original draft:** Norio Kamemura.

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
