## [Decision Letter · Decision Letter 0]

10 Aug 2020

PONE-D-20-12715

Bioactivity of alginetin, a caramelization product of pectin: Cytometric analysis of rat thymic lymphocytes using fluorescent probes

PLOS ONE

Dear Dr. Kamemura,

Thank you for submitting your manuscript to PLOS ONE. After careful consideration, we feel that it has merit but does not fully meet PLOS ONE’s publication criteria as it currently stands. Therefore, we invite you to submit a revised version of the manuscript that addresses the points raised during the review process.

As you can see reviewer #2 suggested minor edits to your manuscript. While reviewer #1 recommended a rejection, I believe his/her comments can be addressed in a revised version. 

We look forward to receiving your revised manuscript.

Kind regards,

Patrick Lajoie, PhD

Academic Editor

PLOS ONE

Journal Requirements:

2. Please add the sources and catalog numbers of all reagents as well as details regarding all equipment used in your study to the Methods section of your manuscript.

In addition, please provide the total number of rats used in your study and clarify the method of euthanasia used.

3. Thank you for including the following funding statement in your acknowledgements section; "The Japan Society for the Promotion of Science (Tokyo, Japan) supports this study with Grant-in-Aids for Scientific Research (C26340039, 18K14408). "

Reviewers' comments:

Reviewer's Responses to Questions

**Comments to the Author**

1. Is the manuscript technically sound, and do the data support the conclusions?

Reviewer #1: No

Reviewer #2: Yes

2. Has the statistical analysis been performed appropriately and rigorously? 

Reviewer #1: I Don't Know

Reviewer #2: Yes

3. Have the authors made all data underlying the findings in their manuscript fully available?

Reviewer #1: No

Reviewer #2: Yes

4. Is the manuscript presented in an intelligible fashion and written in standard English?

Reviewer #1: No

Reviewer #2: Yes

5. Review Comments to the Author

Reviewer #1: In this manuscript, the authors investigated the effects of alginetin on the cell behaviors. I think there are many issues should addressed before published.

1. What is the “pection”?

2. The chemical structure of the alginetin should be investigated. The author state that Alginetin was characterized by 1H-NMR, but no data were provided.

3. The cell morphology images of the cells cultured in the present of alginetin should be provided.

4. the conclusion is too short.

Reviewer #2: Norio Kamemura’s manuscript, named as “Bioactivity of alginetin, a caramelization product of pectin: Cytometric analysis of rat thymic lymphocytes using fluorescent probes”, described the bioactivity algietin as an interesting story, which used fluorescent probes to analyze the zinc level and NPT level in the cytometric analysis. I thought it was good to accept after minor revision:

1. Page 4, line 70 and line 71, the number of H should be identified in the 1HNMR.

2. They didn’t declare clearly the relationship of NPT and oxidative stress.

3. Page 4, line 90 and line 91, it was recommended to use “,” but not “;”.

4. Page 5, line 117, A23187 wasn’t declare well when it was mentioned first time.

5. In the figure 1B, 2, 3B, 5, 7, were these experiments performed once? The error bars weren’t showed in the appropriate way.

6. PLOS authors have the option to publish the peer review history of their article (what does this mean?). If published, this will include your full peer review and any attached files.

Reviewer #1: No

Reviewer #2: No

---

## [Author Response · Author response to Decision Letter 0]

20 Sep 2020

PROS　ONE　

Editors-in-Chief: 

PhD, Patrick Lajoie

Manuscript number: PONE-D-20-12715

Title; Bioactivity of alginetin, a caramelization product of pectin: Cytometric 

analysis of rat thymic lymphocytes using fluorescent probes

Authors: Sayaka Doi, Mina Kawamura, Keisuke Oyama, Tetsuya Akamatsu, Mizuki Mizobuchi, Yasuo Oyama, Toshiya Masuda, Norio Kamemura 

Thank you for your e-mail of 11/Aug/2020. I was pleased to know of your positive evaluation of our manuscript. We have revised the manuscript PONE-D-20-12715 on the basis of the Referee's comments. I logged into the publisher website and submitted the following files related to the revised manuscript: (1) clean version of the manuscript (no mark-up, file name: PONE-D-20-12715-R), (2) marked-up copy showing changes made during the revision (file name: PONE-D-20-12715-R-M), (3) Reply to the comments raised by the reviewer (file name: PONE-D-20-12715-rev), (4) figure files (file names: PONE-D-20-12715-fig)

Appended to this letter is our detailed point-by-point response to the comments raised by reviewer. I agreed with all the comments. 

Thank to your comments, I was able to rewrite my paper more wonderfully. 

We look forward to hearing from you regarding our submission. We would be glad to respond to any further questions and comments that you may have

Sincerely,

Prof. Norio Kamemura

Department of Food-Nutritional Sciences, Faculty of Life Sciences, Tokushima Bunri University Tokushima, Nishihama, Yamashiro-cho, Tokushima, 770-8514, Japan

Tel.: +81 88 602 8095

Fax: +81 88 656 9965

E-mail: kamemura.norio@tks.bunri-u.ac.jp

Journal Requirements:

1. Please ensure that your manuscript meets PLOS ONE's style requirements, 

including those for file naming. 

Reply to the Comment 

We tried to improve our manuscript.

2. Please add the sources and catalog numbers of all reagents as well as details 

regarding all equipment used in your study to the Methods section of your 

manuscript.

In addition, please provide the total number of rats used in your study and 

clarify the method of euthanasia used.

Reply to the Comment 

We tried to improve our manuscript.

Briefly, thymus glands were dissected from 8- to 12-week-old male Wistar rats (Total number of rat; n=8, 280–340 g / Charles River Laboratories Japan, Kanagawa, Japan) anesthetized with 50 mg/kg i.p. thiopental sodium (RavonalTM, Mitsubishi Tanabe Pharma, Osaka, Japan).

3. Thank you for including the following funding statement in your 

acknowledgements section; "The Japan Society for the Promotion of Science 

(Tokyo, Japan) supports this study with Grant-in-Aids for Scientific Research 

(C26340039, 18K14408). "

We note that you have provided funding information that is not currently 

declared in your Funding Statement. However, funding information should not 

appear in the Acknowledgments section or other areas of your manuscript. We will 

only publish funding information present in the Funding Statement section of the 

online submission form.

Please remove any funding-related text from the manuscript and let us know how 

you would like to update your Funding Statement. Currently, your Funding 

Statement reads as follows:

Reply to the Comment 

We tried to improve our manuscript.

Reviewer 1#

We thank referee for careful reading our manuscript and for giving useful comments

1. What is the “pection”?

Reply to the Comment 

We added a new sentence to explain pection.

In the revised manuscript

Abstract (Page 2, Line 24-25)

a naturally-occurring polysacharride found in many plants.

Introduction(Page 3, Line 49-51)

Pectin is a natural produced essential ingredient in preserves. Pectin is a type of starch, called a heteropolysaccharide, that occurs naturally in the cell walls of fruits and vegetables and gives them structure.

2. The chemical structure of the alginetin should be investigated. The author state that Alginetin was characterized by 1H-NMR, but no data were provided. 

Reply to the Comment 

We added a new sentence to explain pection.

In the revised manuscript

Material s and methods (Page 4, Line 75-78)

The structure of the isolated alginetin was characterized by nuclear magnetic resonance spectroscopy (NMR) as follows; 1H-NMR (400 MHz, CD3OD): δ 7.61 (1H, d, J = 7.6 Hz), 7.26 (1H, t, J = 7.6 Hz), 7.19 (1H, d, J = 7.6 Hz), 2.55 (3H, s).

3. The cell morphology images of the cells cultured in the present of alginetin should be provided. 

Reply to the Comment 

We tried to improve our manuscript.

In the revised manuscript

Material s and methods (Page 5, Line 102-104)

The cell morphology images of the cells cultured in the present of alginetin were performed by a inverted microscope (Nikon Eclipse TS100, Nikon,Tokyo, Japan)

Results (Page 6, Line 142-143)

Moreover, alginetin was changed the cell morphology on rat thymic lymphocytes (Figure 1C).

Figure 

Fig 1 (C)

Figure legends (Page 13, Line 317

(C) The cell morphology images in response to 2 h alginetin treatment.

4. the conclusion is too short. 

Reply to the Comment 

We tried to improve our manuscript.

In the revised manuscript

Conclusion (Page 9, Line 216-222)

The biological interaction and toxicity of alginetin has not been studied in detail. Our study showed that alginetin increased cellular content of non-protein thiols and elevated intracellular Zn2+ levels on rat thymic lymphocyte. These results indicate that alginetin increases cell vulnerability to oxidative stress on rat thymocytes. This study provides that alginetin, possessing cytoprotective activity, would provide additional health benefits of jam if it is produced from fruit pectin during jam manufacturing.

Reviewer 2#

We thank referee for careful reading our manuscript and for giving useful comments

1. Page 4, line 70 and line 71, the number of H should be identified in the 1HNMR.

Reply to the Comment 

We tried to improve our manuscript. This NMR data shows that it is algnetin. However, this sentence is written in the paper that has already been submitted.

In the revised manuscript

Material s and methods (Page 4, Line 75-78)

The structure of the isolated alginetin was characterized by nuclear magnetic resonance spectroscopy (NMR) as follows; 1H-NMR (400 MHz, CD3OD): δ 7.61 (1H, d, J = 7.6 Hz), 7.26 (1H, t, J = 7.6 Hz), 7.19 (1H, d, J = 7.6 Hz), 2.55 (3H, s).

2. They didn’t declare clearly the relationship of NPT and oxidative stress.

Reply to the Comment 

We tried to improve our manuscript.

In the revised manuscript

Discussion (Page 8, Line 192-197)

[NTP]i is required to maintain protein thiols in a reduced state and to support a variety of redox reactions for reducing ROS, detoxifying xenobiotics, and facilitating cell signaling. However, excessive oxidative stress results in indiscriminate and irreversible oxidation of protein thiols, depletion of [NTP]i and cell death [14]. Recently, Oyama et al showed that the excessive of [Zn2+]i increase [NPT] i [15]. Our results showed that alginetin elevated intracellular Zn2+ levels increased cellular content of non-protein thiols.

References (Page 12, Line 271-276)

[14] Shahid P. Baba and Aruni Bhatnagar, Role of thiols in oxidative stress. Curr Opin Toxicol. 2018; 7: 133–139.

[15] Akio Kinazaki, Hongqin Chen, Kazuki Koizumi, Takuya Kawanai, Tomohiro M. Oyama, 

Masaya Satoh, Shiro Ishida, Yoshiro Okano, Yasuo Oyama, Putative role of intracellular Zn2+ release during oxidative stress: a trigger to restore cellular thiol content that is decreased by oxidative stress. J Physiol Sci. 2011; 61:403–409.

3. Page 4, line 90 and line 91, it was recommended to use “,” but not “;”.

Reply to the Comment 

We tried to improve our manuscript.

In the revised manuscript

Material s and methods (Page 5, Line 97-98)

(150 mM NaCl, 5 mM KCl, 2 mM CaCl2, 1 mM MgCl2, 5 mM glucose)

4. Page 5, line 117, A23187 wasn’t declare well when it was mentioned first time.

Reply to the Comment 

We tried to improve our manuscript.

In the revised manuscript

Material s and methods (Page 6, Line 125-126)

A23187 (Calcimycin), a divalent cation ionophore (Sigma-Aldrich Co, St. Louis, MO, USA)

5. In the figure 1B, 2, 3B, 5, 7, were these experiments performed once? The 

error bars weren’t showed in the appropriate way.

Reply to the Comment 

Data reported in this study were mean ± standard deviation of 4–8 experiments.

 (127-129). However, we already submited the figure 1B, 2, 3B, 5, 7 which an error bar is written.

In the revised manuscript

Material s and methods (131-134)

Statistical analysis was performed using Excel Toukei 2010 (Social Survey Research Information Co., Ltd. Tokyo, Japan). Statistical analysis was performed using ANOVA with post-hoc Tukey’s multivariate analysis. P-value < 0.05 was considered statistically significant. Data reported in this study were mean ± standard deviation of 4–8 experiments.

---

## [Decision Letter · Decision Letter 1]

13 Oct 2020

Bioactivity of alginetin, a caramelization product of pectin: Cytometric analysis of rat thymic lymphocytes using fluorescent probes

PONE-D-20-12715R1

Dear Dr. Kamemura,

We’re pleased to inform you that your manuscript has been judged scientifically suitable for publication and will be formally accepted for publication once it meets all outstanding technical requirements.

Kind regards,

Patrick Lajoie, PhD

Academic Editor

PLOS ONE

Additional Editor Comments (optional):

Reviewers' comments:

Reviewer's Responses to Questions

**Comments to the Author**

1. If the authors have adequately addressed your comments raised in a previous round of review and you feel that this manuscript is now acceptable for publication, you may indicate that here to bypass the “Comments to the Author” section, enter your conflict of interest statement in the “Confidential to Editor” section, and submit your "Accept" recommendation.

Reviewer #1: All comments have been addressed

Reviewer #2: All comments have been addressed

2. Is the manuscript technically sound, and do the data support the conclusions?

Reviewer #1: Yes

Reviewer #2: Yes

3. Has the statistical analysis been performed appropriately and rigorously? 

Reviewer #1: Yes

Reviewer #2: Yes

4. Have the authors made all data underlying the findings in their manuscript fully available?

Reviewer #1: (No Response)

Reviewer #2: Yes

5. Is the manuscript presented in an intelligible fashion and written in standard English?

Reviewer #1: Yes

Reviewer #2: Yes

6. Review Comments to the Author

Reviewer #1: the authors have revised the manuscript according to the comments of all reviewers, I think it can be accepted.

Reviewer #2: (No Response)

7. PLOS authors have the option to publish the peer review history of their article (what does this mean?). If published, this will include your full peer review and any attached files.

Reviewer #1: No

Reviewer #2: No

---

## [Editor Report · Acceptance letter]

16 Oct 2020

PONE-D-20-12715R1 

Bioactivity of alginetin, a caramelization product of pectin: Cytometric analysis of rat thymic lymphocytes using fluorescent probes 

Dear Dr. Kamemura:

I'm pleased to inform you that your manuscript has been deemed suitable for publication in PLOS ONE. Congratulations! Your manuscript is now with our production department. 

Kind regards, 

on behalf of

Dr. Patrick Lajoie 

Academic Editor

PLOS ONE